# Total Neoadjuvant Therapy for Rectal Cancer: Which Regimens to Use?

**DOI:** 10.3390/cancers16112093

**Published:** 2024-05-31

**Authors:** Kentaro Ochiai, Neal Bhutiani, Atsushi Ikeda, Abhineet Uppal, Michael G. White, Oliver Peacock, Craig A. Messick, Brian K. Bednarski, Yi-Qian Nancy You, John M. Skibber, George J. Chang, Tsuyoshi Konishi

**Affiliations:** 1Department of Colon and Rectal Surgery, The University of Texas M.D. Anderson Cancer Center, Houston, TX 77030, USA; ochiaikun@gmail.com (K.O.); nbhutiani@mdanderson.org (N.B.); aikeda@mdanderson.org (A.I.); auppal1@mdanderson.org (A.U.); opeacock@mdanderson.org (O.P.); cmessick@mdanderson.org (C.A.M.); bkbednarski@mdanderson.org (B.K.B.); ynyou@mdanderson.org (Y.-Q.N.Y.); jskibber@mdanderson.org (J.M.S.); gchang@mdanderson.org (G.J.C.); 2Department of Colon and Rectal Surgery, The University of Tokyo Hospital, Tokyo 113-8655, Japan

**Keywords:** rectal cancer, total neoadjuvant therapy, watch and wait

## Abstract

**Simple Summary:**

Total neoadjuvant therapy (TNT) for rectal cancer is expected to improve oncologic outcomes and organ preservation. Clinicians need to understand the advantages and limitations of the available regimens for multidisciplinary decision making. This article reviews the updated evidence on TNT and addresses tailor-made use of TNT regimens based on tumor location and local and systemic risk.

**Abstract:**

Total neoadjuvant therapy (TNT) is a novel strategy for rectal cancer that administers both (chemo)radiotherapy and systemic chemotherapy before surgery. TNT is expected to improve treatment compliance, tumor regression, organ preservation, and oncologic outcomes. Multiple TNT regimens are currently available with various combinations of the treatments including induction or consolidation chemotherapy, triplet or doublet chemotherapy, and long-course chemoradiotherapy or short-course radiotherapy. Evidence on TNT is rapidly evolving with new data on clinical trials, and no definitive consensus has been established on which regimens to use for improving outcomes. Clinicians need to understand the advantages and limitations of the available regimens for multidisciplinary decision making. This article reviews currently available evidence on TNT for rectal cancer. A decision making flow chart is provided for tailor-made use of TNT regimens based on tumor location and local and systemic risk.

## 1. Introduction

Rectal cancer carries a high risk of developing local recurrence after curative resection, and surgery for rectal cancer is associated with postoperative complications and urinary/sexual dysfunction [1,2,3,4]. To improve therapeutic outcomes, a multidisciplinary approach combining TME with neoadjuvant (chemo)radiotherapy has been developed and widely accepted as a standard of care [5,6,7,8]. However, neoadjuvant (chemo)radiotherapy has not been associated with improvements in overall survival and it delays starting systemic chemotherapy for microscopic metastasis, and there are low rates of completion of adjuvant chemotherapy [9,10,11]. To address these limitations, total neoadjuvant therapy (TNT) has emerged as a novel strategy. TNT involves administration of both (chemo)radiotherapy and systemic chemotherapy before surgery, aiming to improve local and systemic disease control. Potential benefits of TNT include improved tumor response, improved compliance with chemotherapy, improved survival outcomes, and earlier ileostomy reversal. It allows earlier administration of systemic chemotherapy before surgery either before (chemo)radiotherapy (induction chemotherapy) or after (chemo)radiotherapy (consolidation chemotherapy). The reported pathological complete response (pCR) rates after TNT exceed 30%, which is higher than the 15–20% after conventional chemoradiotherapy [12]. Consequently, TNT provides a higher chance for non-operative management (or, watch and wait strategy) after achieving a clinical complete response (cCR) as reported by Habr-Gama et al., and allows more patients to avoid TME-associated surgical complications, anal dysfunction, and ostomy [4,13]. TNT is recommended as a preferred approach for stage II-III rectal cancer in the NCCN guidelines, and is widely adopted in the United States [7]. Although treatment algorithms have been proposed to address which regimens to use in practice, the evidence on TNT is rapidly evolving with new data and long-term follow up of the trials [14] (Table 1). This article aims to review the latest evidence regarding TNT for rectal cancer and provide insights into clinical indications of each TNT regimen.

## 2. Current Evidence on TNT for Rectal Cancer

### 2.1. Induction Chemotherapy before Chemoradiotherapy for Systemic Control

PRODIGE-23

The PRODIGE-23 study compared a TNT regimen consisting of 12 weeks of induction FOLFIRINOX (consisting of fluorouracil, leucovorin, irinotecan, and oxaliplatin) followed by long-course chemoradiotherapy (consisting of 50 Gy radiation over 5 weeks and concurrent capecitabine) and three months of adjuvant chemotherapy, to conventional long-course chemoradiotherapy and six months of adjuvant chemotherapy, targeting patients with cT3/4 M0 rectal cancer [20]. The proportion of cT4 and cN2 disease in the enrolled patients were 18% and 25%, respectively. PRODIGE-23 demonstrated a higher pCR rate of 27.5% with TNT compared to 11.7% with chemoradiotherapy (*p* < 0.0001). The three-year disease-free survival (DFS) rate was improved in the TNT group (75.7% vs. 68.5% with chemoradiotherapy, *p* = 0.034). The study demonstrated a high compliance with the triplet chemotherapy regimen with 92% completing the planned six cycles of FOLFIRINOX, with 45% developing grade 3–4 adverse events. Importantly, recently reported updated long-term follow-up data demonstrated a significant improvement in survival outcomes in the TNT arm compared to the conventional chemoradiotherapy arm. The study demonstrated a 7.6% absolute increase in 5-year disease-free survival, 6.9% absolute increase in 5-year overall survival, 9.9% absolute increase in 5-year distant metastasis-free survival, and 5.7% absolute increase in 5-year cancer-specific survival [19]. This is the first piece of evidence that demonstrated improvement in long-term overall survival with TNT compared to conventional chemoradiotherapy. In light of this improved survival data, a TNT regimen consisting of induction triplet chemotherapy represents an excellent option for fit patients with advanced rectal cancer with high-risk features.

### 2.2. Consolidation Chemotherapy after Chemoradiotherapy for Better Tumor Response

The Timing of Rectal Cancer Response to Chemoradiation Trial

It has been well known that a longer interval from completion of neoadjuvant radiotherapy to surgery increases treatment response and the rate of complete response [24,25]. On the other hand, the longer interval may increase radiation-induced pelvic fibrosis that potentially leads to difficulty of surgery and postoperative complications. Garcia-Agular et al. conducted a well-designed non-randomized phase II trial in clinical stage II–III rectal cancer that compared a standard arm of long-course chemoradiotherapy (50.4 Gy in 28 fractions with concurrent 5-fluorouracil) followed by surgery within 6–8 weeks, with the three TNT arms adding two, four, and six cycles of consolidation FOLFOX (comprising 5-fluorouracil, leucovorin, and oxaliplatin) after chemoradiotherapy before surgery [26,27,28]. The pCR rates increased with each additional course of consolidation FOLFOX: 18%, 25%, 30%, and 38%, respectively (*p* = 0.0036). Despite a higher pelvic fibrosis score in the TNT arms (4.0 out of 10 with TNT vs. 2.4 out of 10 with chemoradiotherapy, *p* = 0.003), there was no significant difference in surgical difficulty (median technical difficulty scale of 4.6, 4.9, 5.1, and 4.8 out of 10, respectively) or postoperative complications between the groups (grade 3 or higher complication of 15%, 15%, 7%, and 5%, respectively, without significant difference). This is an epoch-making study that revealed adding consolidation FOLFOX after chemoradiotherapy with a longer interval to surgery significantly increases complete responses.

CAO/ARO/AIO-12

The CAO/ARO/AIO-12 trial explored the efficacy of consolidation versus induction chemotherapy in TNT for rectal cancer. In this multi-institutional randomized phase II trial, patients with cT3-4 and/or N+ rectal cancer were randomly assigned to either an induction chemotherapy group (three cycles of FOLFOX before 50.4 Gy chemoradiotherapy with concurrent fluorouracil/oxaliplatin) or a consolidation chemotherapy group (three cycles of FOLFOX after chemoradiotherapy). Similar to the Timing of Rectal Cancer Response to Chemoradiation Trial, a longer interval from completion of chemoradiotherapy to surgery in the consolidation group (median 90 days vs. 45 days in the induction group) did not increase postoperative complications. The consolidation group showed a pCR rate of 25% (95% CI, 18% to 32%), which met a predefined statistical hypothesis of an increased pCR rate compared to 15% expected after standard chemoradiotherapy (*p* = 0.001), while the induction group had a pCR rate of 17% (95% CI, 12% to 24%) [21]. The updated report with a median follow-up of 43 months indicated no significant difference between the two groups in 3-year disease-free survival (73% for both groups, *p* = 0.82), 3-year cumulative incidence of locoregional recurrence (6% with induction vs. 5% with consolidation, *p* = 0.67), or distant metastases (18% with induction vs. 16% with consolidation, *p* = 0.52) [29]. A higher pCR rate in the consolidation group suggested that chemoradiotherapy followed by consolidation chemotherapy was a preferred TNT approach to achieve organ preservation.

OPRA

A TNT regimen with a higher rate of complete response provides a higher chance for organ preservation. The OPRA trial was the first large prospective randomized study that investigated outcomes of TNT followed by nonoperative management in patients with excellent response to the therapy [30]. In the OPRA trial, patients with stage II or III rectal cancer were randomly assigned to two TNT treatment arms: the induction chemotherapy arm (FOLFOX/CAPOX for 16–18 weeks followed by CRT for 5.5 weeks) or the consolidation chemotherapy arm (CRT for 5.5 weeks followed by FOLFOX/CAPOX for 16–18 weeks). The outcomes were compared to a historical control treated with conventional chemoradiotherapy. In both arms, patients who demonstrated complete or near-complete response at the restaging evaluation after TNT were offered nonoperative management. The results showed comparable three-year disease-free survival rates between the induction and consolidation arms (76% vs. 76%). These rates were similar to the three year disease-free survival rate among the historical control cohort (75%), indicating no significant improvement in disease-free survival compared to conventional chemoradiotherapy [22]. However, the five-year TME-free survival rates were notably high in both arms; 39% in the induction arm and 54% in the consolidation arm [23]. In addition to the consolidation arm having a higher rate of TME-free survival compared to the induction arm, the regrowth rate after proceeding with nonoperative management was lower in the consolidation arm compared to the induction arm (27% vs. 40%, median F/U three years, *p* = 0.03). In summary, starting with chemoradiotherapy followed by consolidation FOLFOX in the OPRA trial demonstrated a high rate of organ preservation, allowing a substantial number of patients to avoid TME. This regimen is an appealing option for patients desiring organ preservation, particularly in patients with low-lying rectal cancer or those with a threatened radial margin who are at average risk for distant metastases in whom a maximum primary tumor response is desired.

### 2.3. Role of Short-Course Radiotherapy in TNT

Polish II

Neoadjuvant short-course radiotherapy has equivalent oncological benefits in reducing local recurrence compared to neoadjuvant long-course chemoradiotherapy, but the pCR rate and tumor regression after short-course radiotherapy are inferior to chemoradiotherapy [31,32]. Recent phase III trials demonstrated improved pCR rates with TNT using short-course radiotherapy. The Polish II trial compared short-course radiotherapy (25 Gy delivered in five fractions) followed by three cycles of FOLFOX4 (comprising 5-fluorouracil, leucovorin, and oxaliplatin) in the TNT arm, with long-course chemoradiotherapy (50.4 Gy in 28 fractions combined with oxaliplatin and boluses of 5-fluorouracil and leucovorin) in patients with palpably fixed cT3 or cT4 rectal cancer [32]. The trial demonstrated lower acute toxicity in the TNT arm (75% with TNT vs. 83% with CRT, *p* = 0.006) and similar R0 resection rates in both arms (77% with TNT vs. 71% with CRT, *p* = 0.07). The rates of pCR were comparable (16% with TNT vs. 12% with CRT, *p* = 0.17). There were no significant differences in overall survival (49% for both TNT and CRT at eight years, *p* = 0.38) or disease-free survival (43% with TNT vs. 41% with CRT at eight years, *p* = 0.65) between the two groups [16]. Although the trial did not demonstrate the oncological superiority of TNT using short-course radiotherapy compared to conventional long-course chemoradiotherapy, it should be noted that TNT with short-course radiotherapy achieved comparable tumor regression to long-course chemoradiotherapy.

RAPIDO

The RAPIDO trial compared short-course radiotherapy (25 Gy in five fractions) followed by 18 weeks of CAPOX/FOLFOX4 (TNT arm) to a conventional long-course chemoradiotherapy and 24 weeks of adjuvant CAPOX/FOLFOX4 if stipulated by the participating center (CRT arm) [17]. This trial specifically targeted patients with high-risk rectal cancer, including those with cT4 and/or cN2, involved mesorectal fascia, and enlarged lateral lymph nodes. Reflecting this inclusion criteria, the proportion of cT4 and cN2 disease were high, exceeding 30% and 65%, respectively. Patients treated with the RAPIDO regimen demonstrated an improved pCR rate (28.4% with TNT vs. 14.3% with CRT, *p* < 0.0001), lower rates of disease-related treatment failure (23.7% with TNT versus 30.4% with CRT at 3 years, *p* = 0.019), and a lower distant metastasis rate (20.0% with TNT vs. 26.8% with CRT at 3 years, *p* = 0.0048). However, overall survival did not differ between the two groups. Importantly, the results of the long-term follow-up recently reported increased local recurrence in the TNT arm compared to conventional long-course chemoradiotherapy (10.2% with TNT vs. 6.1% with CRT, median follow-up 5.6 years, *p* = 0.027) despite improvement in disease-related treatment failure (27.8% with TNT vs. 34.0% with CRT, *p* = 0.048) and distant metastasis rate (23.0% with TNT vs. 30.4% with CRT, *p* = 0.011). Interestingly, an intraoperative breach of the mesorectum occurred more often in the TNT arm compared to the CRT arm (11% with TNT vs. 6% with CRT, *p* = 0.022), indicating that the quality of TME was more frequently impaired in the TNT arm. Consequently, locoregional recurrence was pronounced in patients with a breached mesorectum (21% with TNT vs. 4% with CRT, *p* = 0.048) [18]. Although the RAPIDO regimen has advantages in shorter treatment duration with reduced distant metastasis and an improved pCR rate compared to long-course chemoradiotherapy, the higher rate of local recurrence without improvement in overall survival may limit its routine use in patients with high-risk rectal cancer.

STELLAR

The STELLAR trial compared short-course radiotherapy (25 Gy in five fractions) followed by four cycles of CAPOX (TNT arm) with conventional long-course chemoradiotherapy (50 Gy in 25 fractions over 5 weeks with concurrent capecitabine) (CRT arm) among patients with T3/4 and/or node-positive low–mid rectal cancer [15]. Postoperative chemotherapy consisted of two cycles of CAPOX in the TNT arm versus six cycles in the CRT arm. The proportion of cT4 and cN2 disease in the enrolled patients was 16% and 34%, respectively. The trial reported that, although grade 3–4 adverse events were more frequent in the TNT group (17.6% with TNT vs. 4.1% with CRT), the compliance rate for neoadjuvant chemotherapy reached 98%. The combined rate of pathological complete response (pCR) and sustained clinical complete response (cCR) was significantly higher in the TNT arm (21.8% with TNT vs. 12.3% with CRT, *p* = 0.002). There were no significant differences in metastasis-free survival (77.1% with TNT vs. 75.3% with CRT, at three years, *p* = 0.475) or locoregional recurrence (8.4% with TNT vs. 11.0% with CRT, at three years, *p* = 0.461). However, the TNT arm had improved overall survival (86.5% with TNT vs. 71.5% with CRT, at three years, *p* = 0.033). Although the improved overall survival observed in the STELLAR trial is noteworthy, the results need careful interpretation given the short follow-up period (median 35 months) and lack of improvement in distant metastasis-free survival or locoregional recurrence.

### 2.4. Induction FOLFOX Aiming Omission of Radiotherapy

PROSPECT

While neoadjuvant radiotherapy contributes to reduced local recurrence and increases the chance of organ preservation, it increases radiation-associated toxicities. Radiotherapy leads to fibrosis and damage to the anal sphincter muscles, resulting in anal dysfunction [33,34]. The PROSPECT trial aimed to investigate whether neoadjuvant chemotherapy (FOLFOX) could replace neoadjuvant chemoradiotherapy [35]. This study evaluated the non-inferiority of neoadjuvant FOLFOX chemotherapy followed by selective chemoradiotherapy only in the non-responders compared to conventional chemoradiotherapy. The study population was patients with stage II–III rectal cancer suitable for anal sphincter preservation, with a low to moderate risk of recurrence (cT2 node-positive, cT3 node-negative, and cT3 node-positive, excluding cT4, four or more enlarged pelvic lymph nodes, and the radial margin <3 mm). Patients who achieved ≥20% response after FOLFOX proceeded with surgery without chemoradiotherapy. Among the 585 patients in the experimental induction FOLFOX arm, 532 patients (91%) were able to omit chemoradiotherapy. The primary endpoint was disease-free survival, which met the non-inferiority criteria of the experimental arm (80.8% vs. 78.6% at five years, a hazard ratio 0.92 (95% CI 0.74–1.14), *p* = 0.005) with a median follow-up of 58 months. This study demonstrated oncologic feasibility of a chemotherapy-alone approach without radiotherapy in patients with low–mid risk rectal cancer planned for sphincter preserving surgery.

### 2.5. Management of Lateral Pelvic Lymph Nodes in the Setting of Neoadjuvant Therapy

The lateral pelvic compartment is a major site of local recurrence after TME in patients with stage II-III low-to-mid rectal cancer due to patterns of lymphatic drainage [36]. The management of lateral pelvic lymph nodes is a topic of active discussion in the era of TNT. In Western countries, LPLND has not been widely adopted as a standard treatment due to a high risk of postoperative complications and uncertain oncological benefits [37]. Instead, emphasis has been placed on neoadjuvant therapy, assuming it can control lateral nodal disease [38]. However, recent studies demonstrated high local recurrence rates after neoadjuvant (chemo)radiotherapy and TME when patients have enlarged lateral lymph nodes. An international multicenter retrospective study of patients with cT3-4 rectal cancer located within 8 cm from the anal verge provided valuable insights on this topic [39]. Patients without an enlarged lateral lymph node (short-axis diameter < 7 mm) had good local control with (chemo)radiotherapy and TME without LPLND with a lateral local recurrence rate of 4.9%. However, patients with an enlarged lateral lymph node (short-axis diameter ≥ 7 mm) had a high local recurrence rate of 25.6% after (chemo)radiotherapy and TME without LPLND, and the majority of local recurrences occurred in the lateral compartment (19.5%). In contrast, patients who received LPLND in addition to TME after (chemo)radiotherapy had a significantly reduced local recurrence rate of 5.7% (*p* = 0.042). These findings suggest that neoadjuvant (chemo)radiotherapy and TME alone is not sufficient to control lateral nodal disease in patients with enlarged lateral lymph nodes, and addition of LPLND may reduce lateral local recurrence.

Based on this data, there is a growing push to combine neoadjuvant treatment and LPLND in patients with an enlarged lateral pelvic lymph node [40,41,42,43]. However, clinical diagnosis of lateral lymph node involvement remains challenging, and there is no clear consensus on diagnostic criteria to select patients who need LPLND. In our single-center study at MD Anderson Cancer Center that evaluated the association between size and histologic positivity of lateral lymph nodes in patients who underwent LPLND after chemoradiotherapy, post-treatment lateral lymph node size ≥5 mm was strongly associated with lateral nodal positivity, suggesting the need of LPLND [38]. In the TNT setting, the data are even more limited. In our recent study that investigated outcomes of patients with rectal cancer treated with TNT, a multidisciplinary MRI-based approach was utilized for deciding whether to perform LPLND [44]. Among 158 patients with stage II–III rectal cancer with a baseline enlarged lateral pelvic lymph node (≥5 mm), 88 patients (56%) underwent LPLND. The decision was made by a multidisciplinary review on MRI findings including pretreatment malignant features (heterogeneity and/or irregularity) and/or posttreatment short-axis diameter ≥ 5 mm. Pathologically positive lateral pelvic lymph nodes were found in 30 cases (34% of patients who underwent dissection). The three-year lateral local recurrence rates were comparable between patients with and without LPLND. These results support the use of MRI-based patient selection that was made for LPLND after TNT at our institution by a multidisciplinary MRI-directed approach.

### 2.6. Molecular Targeted Agent in Total Neoadjuvant Therapy

The role of molecular-targeted agents in neoadjuvant therapy for rectal cancer is currently controversial. In metastatic rectal cancer, the vascular endothelial growth factor inhibitors or epithelial growth factor receptor monoclonal antibodies have shown potential benefits in increasing pCR rate and improving oncologic outcomes. However, the oncologic benefit of using these agents as an adjuvant therapy in non-metastatic rectal cancer remains controversial due to lack of survival advantage [45,46]. The NCCN guidelines currently do not recommend the use of targeted agents in the neoadjuvant settings for non-metastatic rectal cancer given the lack of proven oncologic benefit [7]. A couple of phase II trials have investigated the role of molecular-targeted agents in TNT, expecting increased tumor response by inhibition of tumor proliferation, suppression of angiogenesis and amplification of radiosensitizing effects [47,48].

The EXPART-C trial is a randomized phase II study that investigated the addition of cetuximab to the TNT regimen [49]. The study compared four cycles of CAPOX followed by chemoradiotherapy (50.4 Gy with capecitabine) with versus without weekly cetuximab (CAPOX-C vs. CAPOX). The study population was patients with rectal cancer with high-risk features, including a circumferential resection margin ≤ 1 mm, T3 tumors at or below the levators, extramural extension ≥ 5 mm, T4 classification, or extramural venous invasion. The CAPOX-C arm showed a higher response rate (71% vs. 51%, *p* = 0.028) and improved overall survival (hazard ratio [HR] 0.27, *p* = 0.034). However, pCR rate which was the primary endpoint did not improve with the addition of cetuximab (9% vs. 11%, *p* = 1.0).

The AVACROSS study evaluated a TNT regimen composed of four 21-day cycles of bevacizumab and XELOX, followed by chemoradiotherapy (50.4 Gy in 28 fractions with concurrent bevacizumab and capecitabine) in patients with high-risk rectal cancer [50]. TME was performed 6–8 weeks after chemoradiotherapy. The study reported a high pCR rate of 36%, suggesting potential benefits of bevacizumab in improving tumor response. The INOVA study is another phase II trial that assessed the impact of adding 12 weeks of induction FOLFOX/bevacizumab before chemoradiotherapy/bevacizumab in patients with cT3 low rectal cancer [51,52]. TME was scheduled 6–8 weeks following chemoradiotherapy, similar to the AVACROSS study. The pCR rate was significantly higher in the TNT arm (23.8%) compared to the chemoradiotherapy arm (10%).

Despite such potential oncological benefits of molecular-targeted agents in neoadjuvant therapy, it poses a concern for increased postoperative complications. In the AVACROSS study, 11 patients (24%) required reoperation, half of which were regarding anastomotic leakage. The INOVA study reported postoperative fistulas in seven patients (15.2%) in the TNT arm and nine patients (20%) in the CRT arm with the concurrent use of bevacizumab. The AVACROSS and INOVA studies incorporated concurrent bevacizumab in both induction chemotherapy and consolidation chemoradiotherapy. Although bevacizumab may enhance radiosensitization due to normalization of tumor vasculature, a too-short interval from the use of bevacizumab to surgery may have led to significant postoperative morbidities through impaired wound healing. Given the association between neoadjuvant molecular-targeted agents and wound healing complications, a sufficient interval before surgery may improve outcomes [53]. Konishi et al. reported a phase II study that investigated TNT consisting of induction FOLOFOX plus bevacizumab followed by chemoradiotherapy (50.4 Gy in 28 fractions with concurrent S-1) in patients who had stage II-III rectal cancer with high-risk features (circumferential resection margin ≤ 1 mm, cT4, positive lateral nodes, mesorectal N2 disease, and/or requiring abdominoperineal resection) [54]. In contrast to the AVACROSS and the INOVA study, bevacizumab was not used with chemoradiotherapy. They reported a high pCR rate of 37.2% with acceptable rates of surgical complications including 4.8% anastomotic leakage and 2.4% reoperation. These findings suggest that a longer interval from bevacizumab to surgery may contribute to improved surgical outcomes.

The TRUST trial investigated disease-free survival after six cycles of induction FOLFOXIRI plus bevacizumab followed by chemoradiotherapy (50.4 Gy in 28 fractions with concurrent capecitabine and bevacizumab) in patients with cT4 or T3 rectal cancer [55]. The study reported excellent oncologic outcomes; a 2-year disease-free survival rate of 80.45% and a pCR rate of 36.4%. These prospective studies similarly reported high pCR rates with the use of targeted agents in TNT. A phase III trial is warranted to formally investigate oncologic outcomes of these regimens.

## 3. Treatment Strategy at MD Anderson Cancer Center in 2023–2024

At MD Anderson Cancer Center, we adopt a personalized approach to tailor the use of TNT regimens by local vs. systemic risks, incorporating multiple clinical factors such as tumor height, extramural vascular invasion (EMVI), mesorectal and lateral lymph node involvement, and T4 disease. A typical decision flow chart is presented in Figure 1.

Acknowledging the limitation that this flow chart is based on our institutional interpretation of the evidence, this figure provides an example of how to clinically utilize each TNT regimen. The definitions of high, mid, and low rectum vary across the countries; European guidelines define these terms based on the distance from anal verge [8], whereas American and Japanese guidelines use a classification system based on the location in relation to the pelvic inlet and the anterior peritoneal reflection, with slight differences in their definitions [7,56]. The anatomic relationship between the tumor, peritoneal reflection, and the anorectal junction is crucial in deciding which regimen to use. At MD Anderson Cancer Center, tumor location of the rectum is categorized per the NCCN guideline; high rectal cancer above the peritoneal reflection below the pelvic inlet, mid rectal cancer at the peritoneal reflection, and low rectal cancer below the peritoneal reflection (Figure 2). The typical expected procedure for mid and high rectal cancer is a low anterior resection with the anastomosis above the anorectal junction, whereas the procedure for low rectal cancer is a very low anterior resection with the anastomosis below the anorectal junction or coloanal anastomosis, or abdominoperineal resection.

High-rectal cancer without suspicious lymph nodes or high-risk features carries a low risk of local recurrence and should be managed with upfront surgery given limited benefits from radiotherapy. High rectal cancer with clinically positive lymph nodes can be treated with the PROSPECT regimen, starting with neoadjuvant FOLFOX with the possible omission of radiotherapy [35].

The TNT approach plays a key role in the treatment of mid and low rectal cancers. Decision making regarding the ideal treatment regimen begins with evaluating systemic risk. Given the survival benefit demonstrated by the PRODIGE-23 trial [19], induction triplet chemotherapy followed by chemoradiotherapy is recommended in patients with high-risk features for systemic recurrence (EMVI, significant lymphadenopathy, multiple lateral lymph nodes enlargement, and/or T4 tumor) with reasonable performance status. For low rectal cancer without systemic high-risk features, the regimen employed in the consolidation arm of the OPRA trial (chemoradiotherapy followed by FOLFOX) is preferred to increase the chance of organ preservation [22].

For patients with mid rectal cancer without systemic high-risk features in whom a low anterior resection with anastomosis above the anorectal junction can be performed, the PROSPECT regimen is a reasonable option to avoid radiation-related toxicities. However, in mid rectal cancer with a positive or threatened radial margin (<2 mm), the regimen employed in the consolidation arm of the OPRA trial (chemoradiation followed by FOLFOX is recommended to maximize tumor response.

Given the high local failure rate demonstrated in the RAPIDO trial, we currently employ short-course radiotherapy in TNT selectively for patients with limited access to long-course chemoradiotherapy or those with synchronous metastatic disease necessitating pelvic radiation [18].

Finally, the use of TNT in patients with T2 low-rectal cancer who wish to attempt nonoperative management is controversial, and conventional chemoradiotherapy without TNT remains the standard of care. In this context, the goal of neoadjuvant therapy is to maximize tumor response to achieve a cCR and reduce local regrowth after nonoperative management. While the OPRA trial did not include T2N0 tumors [22], the higher TME-free survival rate with a lower local regrowth rate demonstrated in the consolidation chemotherapy arm of this trial may be extrapolated to justify use of this regimen in T2N0 low rectal tumors when the patient wishes to attempt nonoperative management [33].

## 4. Conclusions

TNT provides improved local control, prolonged survival, and an increased likelihood of organ preservation compared to conventional chemoradiotherapy. The current evidence is limited regarding direct comparisons of each TNT regimen. Clinicians should understand the strengths and limitations of each TNT regimen for clinical use. Acknowledging the limitation that decisions are based on institutional interpretation, multidisciplinary decision making based on local and systemic risk is crucial to tailor the use of TNT regimens to individual patients and their tumors and maximize the benefits of TNT for patient outcomes.

## Figures and Tables

**Figure 1 cancers-16-02093-f001:**
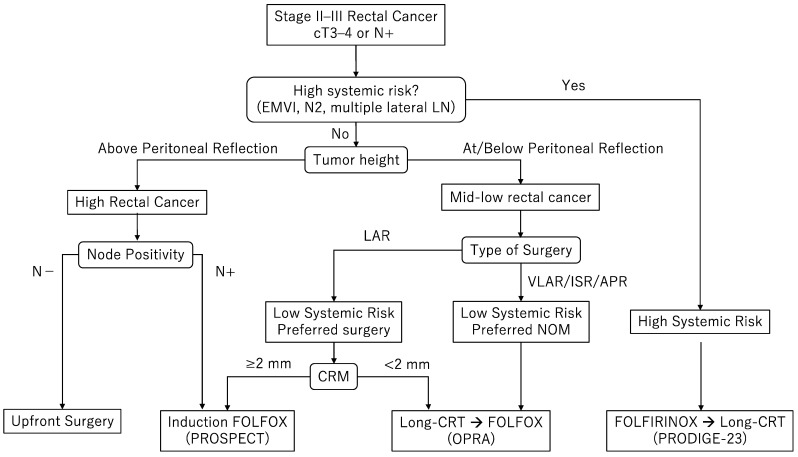
Treatment decision flow chart. EMVI: extramural vascular invasion, LN: lymph node, LAR: low anterior resection, ISR: intersphincteric resection, APR: abdomino-perineal resection, and CRM: circumferential resection margin.

**Figure 2 cancers-16-02093-f002:**
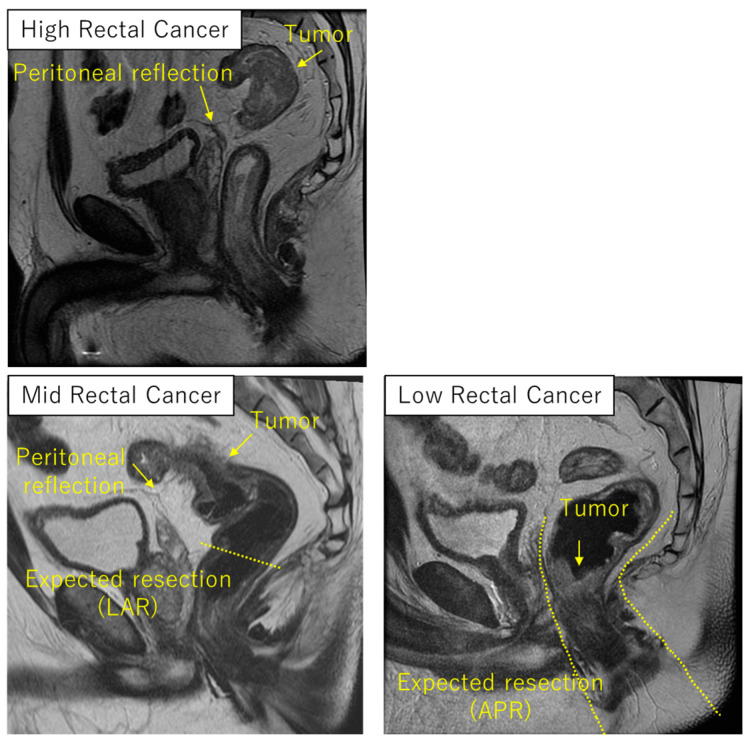
Tumor height according to the relationship between the tumor, peritoneal reflection and the expected procedure. APR: abdomino-perineal resection, and LAR: low anterior resection.

**Table 1 cancers-16-02093-t001:** Trials for total neoadjuvant therapy.

Study	Patients	Arm A	Arm B	pCR Rate	Outcomes
STELLAR [15]	Distal or middle-third rectal cancer stage 3–4 and/or N+ n = 599	Short RT/ Consolidation CAPOX four cycles/adjuvant CAPOX two cycles	Long CRT/adjuvant CAPOX six cycles	26.2% vs. 5.3% (TNT vs. CRT)	3-year DFS 86.5% vs. 75.1%, *p* = 0.033 (TNT vs. CRT)
PolishII [16]	cT4 or fixed cT3 n = 515	Short RT/ Consolidation FOLFOX three cycles	Long CRT	16% vs. 12% (TNT vs. CRT)	8-year DFS 43% vs. 41%, *p* = 0.65 (TNT vs. CRT)
RAPIDO [17,18]	High risk features on MRI (cT4a/b, N2, LLN+, EMVI+, MRF+) n = 920	Short RT/Consolidation CAPOX six cycles or FOLFOX4 nine cycles	Long CRT/adjuvant CAPOX/FOLFOX4 with institutional indication	28.4% vs. 14.3% (TNT vs. CRT)	5.6-year LRF 12% vs. 8%, *p* = 0.07 5.6-year LLR 10% vs. 6%, *p* = 0.027 (TNT vs. CRT)
PRODIGE23 [19,20]	cT3/4 n = 461	Induction FOLFIRINOX six cycles/Long CRT /3 months adjuvant FOLFOX6 or capecitabine	Long CRT/ 6 months adjuvant FOLFOX6 or capecitabine	27.5% vs. 11.7% (TNT vs. CRT)	3-year DFS 76% vs. 69%, *p* = 0.034 7-year OS 81.9% vs. 76.1%, *p* = 0.033 7-year DFS 67.6% vs. 62.5%, *p* = 0.048 (TNT vs. CRT)
CAO/ARO/AIO-12 [21]	Stage II/III rectal cancer n = 306	Induction FOLFOX three cycles/ Long CRT	Long CRT/ Consolidation FOLFOX three cycles	17% vs. 25% (Induction vs. Consolidation)	3-year DFS 73%, *p* = 0.82 (Induction vs. Consolidation)
OPRA [22,23]	Stage II/III n = 324	Induction FOLFOX eight cycles or CAPOX five cycles /Long CRT	Long CRT/ Consolidation FOLFOX eight cycles or CAPOX five cycles	n/a	5-year DFS 71% vs. 69%, *p* = 0.68 5-year TME-free survival 39% vs. 54%, *p* = 0.012 (Induction vs. Consolidation)

CRT: chemoradiation therapy, EMVI: extramural vascular invasion, LLN: lateral pelvic lymph node, LLR: locoregional recurrence, LRF: locoregional failure, MRF: involved mesorectal fascia, pCR: complete pathological response, RT: radiation therapy, TME: total mesorectal exceision, and TNT: total neoadjuvant therapy.

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
