# Peer review of "Total Neoadjuvant Therapy for Rectal Cancer: Which Regimens to Use?"

_cancers, 2024, doi:10.3390/cancers16112093_

Round 1

Reviewer 1 Report

Comments and Suggestions for Authors

The authors report a comprehensive review on TNT for rectal cancer and also provide their institutional decision making flow-chart of TNT regimens based on patient-tumor risk including tumor location and local and systemic risk.  While the manuscript presents a well updated evidences from  phase III trials on TNT, there are some criticisms in these trials to be considered, in particular in the patient selection/elegibility

Patient selection for TNT Trials: I suggest to report more details and comments on patient elegibility/enrolled in phase III trials for a more appropriate intertrial comparison on patient risk (only MRI High-risk patients were eligible in Rapido trial, while more 40% of patients in Prodige-23 and more 50% in Stellar trial were low-intermediate risk)

Limited Generalizability: As the reported decision making flow-chart of TNT regimens is based on authors and institutional interpretation, I suggest  to mention this consideration and to add the limitations of this decision-making flow-chart for its general validity in the conclusions  .

Minor comment: RAPIDO trial compared short-course RT followed by (not following) 18 weeks of CAPOX .. Line 176-177, pg 5.

Comments on the Quality of English Language

The quality of English Language is good, in my opinion

Author Response

Reviewer’s Comment:

Patient selection for TNT Trials: I suggest to report more details and comments on patient elegibility/enrolled in phase III trials for a more appropriate intertrial comparison on patient risk (only MRI High-risk patients were eligible in Rapido trial, while more 40% of patients in Prodige-23 and more 50% in Stellar trial were low-intermediate risk)

Authors’ Response:

We thank the reviewer for the comment. We agree to the reviewer that the characteristics eligible patients differed between trials. The following sentences have been added to the text:

“The proportion of cT4 and cN2 disease of the enrolled patients were 18% and 25%, respectively.” (page 3, lines 69-70, “Current Evidence on TNT for Rectal Cancer” section)

“Reflecting this inclusion criteria, the proportion of cT4 and cN2 disease were high, exceeding 30% and 65%, respectively.” (page 5, lines 182-183, “Current Evidence on TNT for Rectal Cancer” section)

“The proportion of cT4 and cN2 disease of the enrolled patients were 16% and 34%, respectively.” (page 5, lines 208-209, “Current Evidence on TNT for Rectal Cancer” section)

Reviewer’s Comment:

Limited Generalizability: As the reported decision making flow-chart of TNT regimens is based on authors and institutional interpretation, I suggest to mention this consideration and to add the limitations of this decision-making flow-chart for its general validity in the conclusions.

Authors’ Response:

Thanks for the comment. We agree with the comment that the flow chart is our institutional interpretation and might not be applicable in general population. We emphasized that this flow chart is based on our institutional interpretation of the evidence in the text where we introduced Figure 1. We do not think that the conclusion should be changed as we did not mention on this flow chart in the conclusions.

“Acknowledging the limitation that this flow chart is based on our institutional interpretation of the evidence, this figure provides an example on how to clinically utilize each TNT regimen.” (page 9, lines 356-358, “Treatment strategy at MD Anderson Cancer Center in 2023-24” section)

Reviewer’s Comment:

Minor comment: RAPIDO trial compared short-course RT followed by (not following) 18 weeks of CAPOX. Line 176-177, pg 5.

Authors’ Response:

We thank the reviewer for pointing this out. The manuscript has been revised as follows:

The RAPIDO trial compared short-course radiotherapy (25 Gy in five fractions) followed by 18 weeks of CAPOX/FOLFOX4 (TNT arm) to a conventional long-course chemoradiotherapy and 24 weeks of adjuvant CAPOX/FOLFOX4 if stipulated by the participating center (CRT arm)(page 5, lines 177-178, “Current Evidence on TNT for Rectal Cancer” section)

Reviewer 2 Report

Comments and Suggestions for Authors

This is very well written review article addressing a very important topic in the rectal cancer treatment field. The authors also shared their institutional experience which is very helpful for the readers.

No major concerns. 

Only several minor suggestions:

1. Table 1 structure could be improved to make the study arm and control arm easier to be compared. May consider each study list study arm treatment and control arm treatment. 

2. Table 1 Prodige 23 study only listed induction FOLFIRINOX did not mention the adjuvant chemo component in the study arm. 

3. Table 1 OPRA study may consider using the updated 5-year data instead of the 3-year data. 

4. Under the section "current evidence on TNT for rectal cancer", within the PRODIGE 23 paragraph, the control arm in this study should be conventional long course chemoradiation and 6 months adjuvant chemotherapy. The authors should change the 3 months to 6 months. 

5. Under the section treatment strategy at MD Anderson Cancer Center in 2003-24, the authors reported short course radiation in TNT was not used which is reasonable given the higher local failure risk. However, limiting its use in general is possibly overreacting. Using RAPIDO regimen but change the sequence of treatment (giving the short course radiation after induction chemotherapy immediately before surgery for patients whose disease responded to induction chemo well) seemed to be a reasonable approach. 

6. The definition of upper rectum (named high rectum) could be different, above peritoneal reflection is used here. Although this is a reasonable definition, need to be used carefully to avoid overtreatment for patient has distal sigmoid colon cancer (since per this paper figure 1, these patient with suspicious lymph node all had neoadjuvant chemo. 

Author Response

Reviewer’s Comment:

  1. Table 1 structure could be improved to make the study arm and control arm easier to be compared. May consider each study list study arm treatment and control arm treatment.
  2. Table 1 Prodige 23 study only listed induction FOLFIRINOX did not mention the adjuvant chemo component in the study arm.

Authors’ Response:

We thank the reviewer for these comments. The table 1 has been revised as attached:

Reviewer’s Comment:

  1. Table 1 OPRA study may consider using the updated 5-year data instead of the 3-year data.

Authors’ Response:

We thank the reviewer for the comment. The table 1 has been revised. Manuscript has also been revised to reflect the updated results of the OPRA trial:

“However, the five-year TME-free survival rates were notably high in both arms; 39% in the induction arm and 54% in the consolidation arm”. (page 4, lines 143-144, “Current Evidence on TNT for Rectal Cancer” section)

Reviewer’s Comment:

  1. Under the section "current evidence on TNT for rectal cancer", within the PRODIGE 23 paragraph, the control arm in this study should be conventional long course chemoradiation and 6 months adjuvant chemotherapy. The authors should change the 3 months to 6 months.

Authors’ Response:

We thank the reviewer for pointing this out. The manuscript has been revised as follows:

“The PRODIGE-23 study compared a TNT regimen consisting of 12 weeks of induction FOLFIRINOX (consisted of fluorouracil, leucovorin, irinotecan, and oxaliplatin) followed by long-course chemoradiotherapy (consisted of 50 Gy radiation over 5 weeks and concurrent capecitabine) and three months adjuvant chemotherapy, to conventional long-course chemoradiotherapy and six months adjuvant chemotherapy, targeting patients with cT3/4 M0 rectal cancer.” (page 3, lines 64-69, “Current Evidence on TNT for Rectal Cancer” section)

Reviewer’s Comment:

  1. Under the section treatment strategy at MD Anderson Cancer Center in 2003-24, the authors reported short course radiation in TNT was not used which is reasonable given the higher local failure risk. However, limiting its use in general is possibly overreacting. Using RAPIDO regimen but change the sequence of treatment (giving the short course radiation after induction chemotherapy immediately before surgery for patients whose disease responded to induction chemo well) seemed to be a reasonable approach.

Authors’ Response:

We appreciate the reviewer's comment. While we do not routinely employ the RAPIDO regimen, we employ short-course radiotherapy in TNT selectively for patients with limited access to long-course chemoradiotherapy or those with synchronous metastatic disease necessitating pelvic radiation. We have revised the manuscript to reflect this:

Given a high local failure rate demonstrated in the RAPIDO trial, we currently employ short-course radiotherapy in TNT selectively for patients with limited access to long-course chemoradiotherapy or those with synchronous metastatic disease necessitating pelvic radiation.” (page 10, lines 396-399, “Treatment strategy at MD Anderson Cancer Center in 2023-24” section)

Reviewer’s Comment:

  1. The definition of upper rectum (named high rectum) could be different, above peritoneal reflection is used here. Although this is a reasonable definition, need to be used carefully to avoid overtreatment for patient has distal sigmoid colon cancer (since per this paper figure 1, these patient with suspicious lymph node all had neoadjuvant chemo.

Authors’ Response:

As the reviewer has pointed out, the definition of the rectum varies among European, NCCN, and Japanese guidelines. In this article, the definition of the rectum was based on the NCCN guideline, ie, below the pelvic inlet above the anterior peritoneal reflection. We revised the manuscript as follows:

The definitions of high-, mid-, and low-rectum vary across the countries; European guidelines define these terms based on the distance from anal verge, whereas American and Japanese guidelines use a classification system based on the location in relation to the pelvic inlet and the anterior peritoneal reflection, with slight differences in their definitions.” (page 9, lines 360-361, “Treatment strategy at MD Anderson Cancer Center in 2023-24” section)
“At MD Anderson Cancer Center, tumor location of the rectum is categorized per the NCCN guideline; high rectal cancer above the peritoneal reflection below the pelvic inlet, mid rectal cancer at the peritoneal reflection, and low rectal cancer below the peritoneal reflection (Figure 2).” (page 9, lines 364-365, “Treatment strategy at MD Anderson Cancer Center in 2023-24” section)

Round 2

Reviewer 1 Report

Comments and Suggestions for Authors

The authors improved appropriately their manuscript. I congratulate for this interesting contribution in the management of patients with rectal cancer

Comments on the Quality of English Language

The quality of English Language is good, in my opinion